# Human-Animal Interaction in Animal-Assisted Interventions (AAI)s: Zoonosis Risks, Benefits, and Future Directions—A One Health Approach

**DOI:** 10.3390/ani13101592

**Published:** 2023-05-09

**Authors:** Giovanna Liguori, Anna Costagliola, Renato Lombardi, Orlando Paciello, Antonio Giordano

**Affiliations:** 1Department of Veterinary Medicine and Animal Productions, University of Napoli Federico II, 80138 Napoli, Italy; giovanna.liguori@unina.it (G.L.); orlando.paciello@unina.it (O.P.); 2Local Health Authority, ASL Foggia, 71122 Foggia, Italy; renato.lombardi@aslfg.it; 3Sbarro Sbarro Institute for Cancer Research and Molecular Medicine, Center for Biotechnology, College of Science and Technology, Temple University, Philadelphia, PA 19122, USA; giordano@temple.edu; 4Department of Medical Biotechnology, University of Siena, 53100 Siena, Italy

**Keywords:** animal-assisted interventions, zoonosis, one health, animal welfare, preventive medicine

## Abstract

**Simple Summary:**

Animal-assisted interventions (AAI)s are planned activities carried out in multidisciplinary teams with educational, therapeutic, and ludic-recreational purposes. The multidisciplinary and integrated character identifies AAIs as the expression of one health. While AAIs offer many advantages to subjects, they could be exposed to several zoonotic-pathogens transmissions. Therefore, positive animal welfare, as preventive medicine to avoid accidents or zoonotic transmissions, is a relevant aspect with implications for human and animal health and welfare. The knowledge of several pathogens causing zoonoses in the animal species employed during the AAIs, as well as the preventive measures aimed at reducing and/or preventing the risk, guarantee their safety for patients. This review defines the future outcomes of the AAIs including the strengthening of preventive measures according to a “One Health approach”, the training of the personnel involved in AAIs, and the standardized protocols for hygiene–health–behavioral monitoring, aimed at a future production of health and behavioral certifications.

**Abstract:**

Animal-assisted interventions (AAI)s represent the expression of integrated medicine, according to the One Health approach. Actually, animal-assisted therapies and animal-assisted activities are implemented in hospitals, rehabilitation centers, etc. The efficacy of AAIs is based on interspecific interactions and would be impacted by different factors, such as the characters of both the animal and the handler, a suitable selection of animal species, an appropriate animal educational protocol, the relationship between the handler and the animal, and mutual relationship among the animal, the patients, and members of the working team. AAIs produce many advantages for the patients but could expose them to zoonotic-pathogens transmission. Therefore, positive animal welfare, as preventative medicine to avoid incidents or transmission of zoonosis, is a relevant aspect with implications for human and animal health and wellbeing. This review aims to summarize the current published knowledge regarding the occurrence of pathogens in AAIs and to discuss their relevance in light of health and safety in AAIs participants. In addition, this review will contribute to defining the state of the art of AAIs through a careful benefits/challenges analysis and offers discussion points on the possible future developments according to the One Health approach.

## 1. Introduction

Animal-assisted interventions (AAI)s, generally defined as ‘pet therapy’, represent a wide variety of programmed activities involving animals finalized to the improvement of human health. AAIs are characterized by a multidisciplinary staff of human beings and animals with the purpose of striving for therapeutic (animal-assisted therapy—AAT) or educational (animal-assisted education—AAE) outcomes. In addition, AAIs involve animal-assisted activities (AAA) and AAI resident animals (RA) [1,2], as well as nonformal interventions performed on a voluntary basis by the multidisciplinary team with motivational, educational, or recreational goals [1]. AATs contribute to improving the health and welfare of the patients [3] and can play an adjuvant role during emotional–affective, social, linguistic, and cognitive rehabilitation [4,5,6,7]. Therefore, several scientific studies, mainly concerning dog-assisted therapy, led researchers to highlight the substantial beneficial effects induced in patients affected by depression/anxiety and/or psychophysical disorders, such as adults with autism spectrum [8], or Alzheimer’s diseases and other forms of dementia [6,9,10], or during psychotherapy sessions for adolescents [11], and in post-traumatic stress disorders in adults, adolescents, children, and soldiers [11,12,13,14].

The constraints of some pharmacological therapies against Alzheimer’s disease, autism spectrum disorder, and Parkinson’s disease, have conducted researchers to search for new therapeutic options, focusing on AATs. These are considered therapies to complement conventional ones [4,5,6,8,15,16]. However, the new frontier of integrated medicine focuses on the prevention and treatment of several diseases via a biopsychosocial multidisciplinary approach [17,18,19,20].

AAIs are now introduced in several different settings, including schools, nursing homes, hospitals, prisons, daycare centers, and social farms [21,22,23], even though more evidenced prominent research is still necessary [21,24]. The public great uproar recently encountered about the topic has ordered the sector to consider the role of animals in human health promotion, and focus on ethical [25], safety [21,26], and economic debates [21,27].

Thus, some international associations and organizations have elaborated standards and best practices for AAIs such as the White Paper of the International Association of Human-Animal Interaction Organizations [28], or the Animal-Assisted Interventions Code of Practice for the UK, edited by the Society for Companion Animal Studies [29]. Similar to other Western countries, in 2015, an agreement between the Italian Government, the regional authorities, and the autonomous provinces of Trento and Bolzano was established, setting up guidelines on AAIs [30] with the aim of guiding and improving the development of the AAI topic through a dialogue among the institutions, the stakeholders, and the scientific world.

Currently, all Italian regions are transposing and implementing the agreement. In the Italian Agreement, five animal species are cited: the dog (*Canis familiaris*), the cat (*Felis catus*), the rabbit (*Oryctolagus cuniculus*), the horse (*Equus caballus*), and the donkey (*Equus asinus*) [21].

Although most of the studies on AAI have focused on dog-assisted interventions, equine-assisted interventions (EAI) as an emerging field of AAI have been found to promote physical and mental health. In EAI, horses also cofacilitate therapeutic or recreative environments. EAI includes several activities like horseback riding, but also activities such as adapted acrobatics, carriage driving, grooming, and caring for and interacting with horses [31]. Since EAI seems to alleviate emotional, social, cognitive, and physical disorders in various healthcare settings, this procedure is currently used to treat neurological or psychiatric disorders [32,33,34] and other disabilities such as autism-spectrum disorder [35,36]. In addition, EAI positively affects people affected by Down’s syndrome [37] or poststroke subjects [38]. During EAI, the rider’s movements are rhythmic and regular [39]; therefore, patients affected by motor disabilities tend to adapt their locomotor system to the rhythmic movements of the horse to improve balance and coordination [40,41,42].

In addition to dogs, cats, rabbits, horses, and donkeys, which can be involved in AAT and AAE, there are other domestic animal species that are involved in AAA as well (guinea pigs, chickens, goats, ferrets, etc.), although they still need to be evaluated by national authorities as regards their safety and welfare before they can become suitable officially for AAT and AAE [21]. In other countries, AAIs involve, in addition to domesticated animals, more exotic species (e.g., dolphin) or others such as alpaca [43,44,45], as well as small mammals (e.g., guinea pig), employed for improving the physical and mental functioning of both children and adults [46].

The “co-therapies” implies the physical contact between the patients [47] and the animals’ mucosae and fur that cause the exposition of several pathogens carried by animals in a silent way [48,49,50]. Therefore, noteworthy is the dispute about an existing necessity to maintain physical contact during AAI and prevent the transmission of potential zoonotic agents. Zoonosis is an infectious disease that can be transmitted to humans from animals and vice versa [51].

The present narrative review focuses on the most common zoonotic agents that can be transmitted from animals to humans and the ways of preventing them, to be applied both to the animals and to the patient; in choosing the right animal species according to the single patient’s needs and link that might occur during AAIs, according to the Italian National Guidelines for AAIs 2015 [30]. For this purpose, these guidelines include (a) specific AAI training for each figure involved in AAI (veterinarian, animal handler, etc.); (b) the establishment of a regional register of traders and facilities; (c) health, welfare, and behavioral criteria for the animals involved; and (d) the scientific evaluation of the results of projects performed [21]. The narrative review also intends to focus on the benefits/challenges of AAIs for this purpose. Four electronic scientific databases have been consulted: PubMed, Web of Science, SCOPUS, and Google Scholar. The search of documents for this narrative review was performed in January 2023, by two of the authors (GL and AC) independently, using five strings that represent subject heading and text words related to our topic (see: Table 1).

## 2. Potential Zoonoses Associated with Dogs

The dog represents the mainly involved animal species in AAIs as well as in hospital-based AAT in pediatric oncology patients [47]. Although substantial benefits have been documented in the literature on the involvement of dogs in AAI, physical contact between humans and animals could cause the carrying of potential pathogens from animals to human beings. Sometimes, in AAI settings the dogs employed might also represent pathogen carriers causing a pathogen exchange from being contaminated by physical contact with infected/haunted subjects and then transferring microorganisms to other patients [2,52]. In particular, it is scientifically demonstrated that pediatric oncological patients have immunodeficiency cancer treatments mediated and therefore are retained to be more susceptible [2,47]. The modes of pathogens transmission can include physical interaction, aerosol inhalation, infected saliva, urine, or feces, as well as contact with contaminated tools [53].

Several transmissible diseases that can be transmitted by dogs to patients (see Table 2) consist of viral pathogens (Norovirus, rabies) [54], microorganisms of bacterial origin such as Bordetella bronchiseptica-associated disease, Brucellosis, *Capnocytophagosis, Coxiellosis, Cryptosporidiosis, Infections with E. coli, Leptospirosis, Methicillin resistance Staphylococcus aureus*, *MRSA*, including *Campylobacter* spp, such as *C. jejuni*, and *C. coli* [55]. The most common risk factors associated with *Campylobacter* spp. occurrence are represented by the dog’s breed and diet, while differences in age and sex of the animal do not seem to have statistical significance.

Considering that younger dogs have an immature immune system and a poorly developed gut microbiota, therefore unable to induce the competitive exclusion principle toward microorganisms, *Campylobacter* spp. incidence is more easily found in the latter [56]. In addition, purebred dogs seem to be more easily exposed to *Campylobacter* spp. coinfection if compared to crossbreed dogs, which appear generally more resistant to disease [56]. Another potential risk for dogs is *Campylobacter* spp. transmission from homemade cooked food, especially meat, and is sometimes associated with incorrect food handling and/or cross-contamination from raw foods [55,57]. Dogs involved in AAIs should not be fed raw animal-origin foods within 90 days before service [58,59]

Still, other pathogens that cause zoonoses and could be found in dogs are Pasterurellosis, Salmonellosis, Staphylococcal pyoderma, Tularemia cutaneus, Yersiniosis enterocolitica, fungal contamination, ringworms, parasites (Echinococcosis, Giardiasis, Mange), and visceral larva migrans [2,53,60,61].

The adoption of different practices and protocols can modulate safety results and outcomes [2,62].

Protocols to evaluate zoonotic-pathogens transmission consist of group A streptococci, *Clostridium difficile*, vancomycin-resistant enterococci, and MRSA [58]. The majority of zoonoses are commonly identified and treated. Preventive measures involve internal and external parasite treatments and vaccination protocols; taken together, they are environmental biomonitoring to safeguard both dogs’ and human health [2,53]. An Italian study showed that the most common high-risk zoonotic agents of parasitic origin included Ancylostomatidae, *Eucoleus aerophylus*, *Toxocara canis*, *Giardia duodenalis*, *Nannizzia gypsea*, and *Paraphyton mirabile* [63].

The risk of pathogen transmission depends on contact and time of exposure to the pathogens; therefore, the duration of an AAT session varies from 15 to 120 min, with increasing intensification of the physical contact between the animal and the patient [4,5,6,64].

Another problem that can be found in AAIs is the patient’s or family’s fear of dogs; one of the exclusion criteria of an animal involved in AAI in such circumstances. In fact, the exclusion criteria can include fear/phobia of animals, cultural attitudes [65]; unsafe animal behavior [66]; injuries, such as a fall or bites and scratches, which can happen if handled inappropriately or an inappropriate animal for the therapy environment is chosen [25,67]; allergic reactions such as pet dander [68]; as well as concerns regarding hygiene/sanitization [68].

Most protocols suggest that dogs undergo washing and grooming by 24 h prior to contact with patients [69]. Nevertheless, washing a dog a couple of times during the week can reduce *Can f1* from dog hair and coat, lowering the risk of allergic reaction [25]. Allergies can cause skin irritations, rhinoconjunctivitis, and bronchial asthma, representing a precise contraindication for (in)direct contact with dogs [25]. Up to now, there is no evidence of the presence of a hypoallergenic dog breed [68].

Then, from a welfare point of view, the allergological risks of implementing dogs in a hospital or outpatient setting should not be underestimated [68], and these animals should not interact with patients suffering from allergic asthma. In addition, emergency medication should be available to a trained person on site to mitigate risk in the event of an allergic response [68,70].

**Table 2 animals-13-01592-t002:** Potential pathogens and the preventive measures to reduce/eliminate the zoonotic risk in AAIs.

Animal Species	Zoonotic Agents	References	Preventive Measures
Dog	**Viruses:** Norovirus Lyssavirus: Rabies virus **Bacterial infections:** Bordetella bronchiseptica-associated disease Brucellosis, Campylobacteriosis, Capnocytophagosis, Coxiellosis, Cryptosporidiosis, *E. coli*, Leptospirosis, Methicillin resistance Staphylococcus aureus (MRSA), *Campylobacter jejuni,* and *C. coli* Group A Streptococci, Clostridium difficile, vancomycin-resistant enterococci, Pasterurellosis, Salmonellosis, Staphylococcal pyoderma, Tularemia cutaneus, Yersiniosis enterocolitica Fungal infections: ringworms; Parasites: Echinococcosis, Mange visceral larva migrans Ancylostomatidae, *Eucoleus aerophylus*, *Toxocara canis*, *Giardia duodenalis*, *Nannizzia gypsea*, *Paraphyton mirabile*	[38,54] [58] [2,53,60,61] [63]	Basic sanitation practices Internal and external parasites treatments Vaccinations Environmental control [2,53]
Cat	**Viruses:** cowpox, rabies **Bacteria:** *Bacillus anthracis*, Bartonella species, *Borrelia burgdorferi*, group A streptococcus, *Listeria monocytogenes*, *Rickettsia felis,* salmonella species, **Tapeworms:** *Dipylidium caninum*, *Echinococcus multilocularis*, **Ectoparasites:** *Cheyletiella blakei*, *Sarcoptes scabiei*, **Roundworms:** *Ancylostoma braziliense*, Hearthworm: *Dirofilaria immitis*, *Strongyloides stercoralis*, *Uncinaria stenocephala*; **Fungi:** Microsporum species, Trichophyton species; Protozoans: *Toxoplasma gondii*	[71] [72]	Basic sanitation practises Vaccination Regular testings Deworming Good hygiene [71] Vaccination long-term safety of an anti-Fel d 1 IgY-supplemented diet [72]
Rabbit	**Bacteria:***Pasteurella multocida* **Protozoa:***cryptosporidium* **External parasites:** Cheyletiella (acariasis), Dermatophytosis Trichophyton	[73] [74,75]	Best sanitation practices [73]
Horse	**Viruses:** Borna Disease Virus-1 Alphavirus, Hendravirus, Lyssavirus Group-A-Rotavirus, Vesiculovirus Flavivirus **Bacteria:** *Anaplasma phagocytophilum* *Bacillus antracis, Clostridium botulinum* *Brucella abortus and B suis* *Clostridium difficile, Burgholderia mallei, Leptospira interrogans*, *Borrelia burgdorferi, Strains of Staphylococcus aureus, Rhodococcus equi, salmonella enterica ssp enterica serovar typhimurium, Streptococcus equi ssp zooepidemicus, Clostridium tetani, Mycobacteria avium, bovis, and tuberculosis*, **Protozoa:** *Cryptosporidium parvum, Giardia intestinalis, Toxoplasma gondii, Trichinella* **Fungi:** *Microsporum canis, Micosporum gypseum, Trichophyton mentagrophytes, Trichophyton equinum*	[76]	Basic sanitation practices Healthy environment, vaccines [76]
Donkey	**Viruses:** Equine encephalomyelitis, Rabies, **Protozoa:** Toxoplasma **Tapeworm:** Hydatidosis,	[77]	Best sanitation practices Regular anti ecto- and endo-zoonotic parasites treatments, Personal and environmental hygiene, regular dermatophytosis monitor [63]

## 3. Potential Zoonoses Associated with Cat-Assisted Interventions

Potential zoonotic diseases from cats that can infect humans are listed in Table 2. The most common zoonotic diseases that can be transmitted by cats include *Bacillus anthracis*, Bartonella species, *Borrelia burgdorferi*, group A streptococcus, *Listeria monocytogenes*, *Rickettsia felis,* salmonella species, tapeworms (*Dipylidium caninum*, *Echinococcus multilocularis*), ectoparasites (*Cheyletiella blakei*, *Sarcoptes scabiei*), Roundworms (*Ancylostoma braziliense*), heartworm (*Dirofilaria immitis*, *Strongyloides stercoralis*, *Uncinaria stenocephala*); fungi (Microsporum species, Trichophyton species); protozoans (*Toxoplasma gondii*); viruses (cowpox, rabies). Some cat-associated zoonoses, such as rabies, are prevented through vaccination. Other infections can be prevented or eliminated through regular testing and the deworming of the animals by veterinarians. In addition, good hygiene should always be maintained around animals.

Another aspect of involving cats is the expression of allergic symptoms from the patients or their relatives. Allergy to cat fur depends on the presence and spread out of the Fel D1 major antigen found in their saliva and dander [72]. This allergen is produced more by entire male cats than neutered males (due to a partial influence from testosterone) [78,79,80,81,82] and females. No scientific data can confirm the presence of less allergenic breeds of cats [72]. Two strategies to reduce the problem could be summarized as follows: 1. Vaccination of cats against the allergen Fel d 1 secretion, not commercially available yet; 2. Long-term safety of an anti-Fel d 1 IgY-supplemented diet, available commercially. The availability of this diet helps the veterinary healthcare team to organize, with both owners and human healthcare professionals, measures that can be taken to reduce the environmental burden of Fel d 1 [72] or in extreme cases choosing a different animal species for the therapy.

An integrated approach targeting the mechanisms causing allergy by dog or cat include allergenic immunotherapy (AIT), subcutaneous-(SCIT) or sublingual-(SLIT) immunotherapy, patient education, allergen avoidance, and pharmacotherapy.

## 4. Potential Zoonoses Associated with Rabbit-Assisted Interventions

The literature referring to this species is almost scarce and only a few data were possible to be extracted (see Table 2). Rabbits are generally docile animals that are easy to handle, with a low risk of transmitting zoonotic pathogens [73]. The greater risk when working with rabbits is to develop allergies while biting is uncommon. Specific handling techniques and appropriate protective clothes are necessary to avoid painful scratches with their rear limbs when improperly handled.

Diseases of public health importance in domestic rabbits are rare. The development of disease in the human host usually occurs in the presence of a preexisting compromised immune system. Potential rabbit zoonoses include *Pasteurella multocida* (the bacterium that lives in the oral cavity or upper respiratory tract of rabbits and can be transferred to humans by bites or scratch) [73], Cryptosporidiosis, induced by *Cryptosporidium*, an extracellular protozoal organism, which is transmitted via the fecal–oral route [74,75]. Other rabbit diseases such as salmonellosis, yersiniosis, and tularemia are rare and can be transmitted to humans by wild rabbits. External parasites such as acariasis (Cheyletiella) and dermatophytosis (Trichophyton) may be transmitted to humans [75].

## 5. Potential Zoonoses Associated with Equine-Assisted Interventions

Several infectious diseases can affect both horses and humans (see Table 2). Zoonotic agents represent the interconnection between human beings, the environment, and animals [83], thus, their transmission from horse to human can occur through direct contact (e.g., Hendra virus) or indirect contamination such as food products (e.g., Botulism) or vectors like ticks (e.g., Lyme-Borrelioses) and mosquitos (e.g., West Nile Fever) [84]. Therefore, the incidence of zoonotic-pathogens transmission has a close relation to the immune system of the animal and human beings, pathogen-control protocols, and environmental influence [85]. The frequency of zoonotic diseases is strictly under control [83]. No findings have been found on reverse zoonosis, which is a disease that was transmitted from human to horse.

Novel pathogens seem to have a relevant impact on horse/human health and relationships. The most emblematic case of emerging disease is the Hendra virus (HeV), which was considered the leading cause of mortality in both human beings and horses in Australia. This virus is a part of the paramyxoviridae genus, which has as a natural reservoir the flying foxes. The transmission way of the above virus is the direct contact of people with the secretions of contaminated horses [86]. Middleton and coworkers (2014) [87] developed an anti-HeV vaccine for horses that provided health benefits to humans as well as to environmental health too, meeting the spirit of a One Health approach. Recently, the pandemic of coronavirus disease 2019 (COVID-19) has triggered possible questions regarding the involvement of horses as virus reservoirs. In the USA it was described that about 10% of horses were positive for β-coronavirus [88], one of the potential sources of COVID-19. At present, no scientific evidence of a direct relationship between human coronavirus and that of horses has been reported [52]. No literature has been detected regarding the direct transmission of equine-human zoonotic agents during EAI. The public should be instructed and trained before interacting with the horse; furthermore, legislation with formal and social permission to work with the horse should be necessary. While integrated medicine is now related to zoonosis prevention, other One Health aspects are important, such as pathogen-control protocols and manufacturing, safety, and quality control of vaccines as well [76].

Among equine species, donkeys are also employed in assisted therapy in elderly people with mental disorders [89,90] or children with autism spectrum or motor disabilities [91,92]. These animals seem to be excellent facilitators in the motivation-building process, being able to stimulate the child’s development promoting psychoaffective and psychocognitive development processes [91]. The most important selective feature of donkeys for AT is their temperament evaluated by numerous behavioral tests used in horses too [93]. Nevertheless, knowledge of donkeys’ responses to standardized behavioral tests should be improved [94]. As reported by De Rose et al. (2011) [91], the choice of the donkey is based on its ethological characteristics, primarily its size, which makes the donkey an unavoidable but not intimidating interlocutor. Thanks to its physical structure, the donkey is physically well-accepted, offering an opportunity for contact and space sharing. When confronted with a new situation, the donkey is neither impulsive nor anxious and instinctively curious [94,95].

No literature regarding zoonotic agents’ transmission from donkeys during AAIs has been found, with the exception of Hydatidosis and toxoplasma infections where their zoonotic potentials have been studied more. Bacterial zoonoses such as brucellosis, leptospirosis, and salmonellosis have been described mainly in equine. In contrast, rabies and Equine encephalomyelitis viral infection have been reported in donkeys more than any other viral zoonoses [77].

The prevention and control measures put in place and the education of the public about them represent the way to prevent possible human infections.

## 6. Discussion

The human–animal bond existed for thousands of years and such a relationship is important for veterinary medicine and human health and wellbeing [96,97]. Veterinarians’ involvement in AAIs is crucial because they serve for the health and welfare of animals participating in these programs and as experts in zoonotic disease transmission. Nevertheless, AAIs might be regulated by basic procedures, be closely supervised, and have adequately trained operators. The health and welfare conditions of the human beings and animals involved should be guaranteed.

Since veterinary medicine serves society, it fulfills both human and animal needs. Veterinarians, as professionals, are qualified to provide community service via such programs and to aid in the scientific evaluation and documentation of the health benefits or risks of animal-assisted interventions [96]. Involving animals in AAIs has become a common extensive practice. However, different practices and protocols used across institutions could potentially compromise the outcomes, safety, and findings [2]. An Animal Assisted Intervention International Standards of Practice [98] help and encourage individuals, organizations, institutions, and health and human service providers who are interested in or are organizing an AAI program. These standards represent the minimum necessary to conduct an AAI program for Animal Assisted Intervention International (AAII) members. These programs should also meet any standards or regulations requested by governing bodies for their region and their home organization [98]. Such AAII standards of practice refer to dogs but can be extended and applied to all different animal species employed in AAIs. Despite these benefits, AAIs also show problems, such as allergies, asthma, animal bites and scratches, and human falls. Bites, slips, and falls from animals, dogs, and, to a lesser extent, cats might be the most troublesome animal-associated health hazard [99] in terms of seriousness, frequency, and cost. Moreover, for the most troublesome breeds, the importance of good temperament and the need for schooling could be incorporated into guidelines to reduce the risk of injuries in an AAI program. It is reasonable to suggest that, in a well-supervised environment such as a hospital ward, after careful selection of the AAI animal involved, the risks of animal bites are minimal and should not prevent the implementation of such therapy. In fact, animal-related accidents have to be considered very rare although they can happen. For that, appropriate reviews and guidelines implement all security precautions effectively minimizing risks [25,67].

One of the most important problems regarding AAIs is the fear of transmission of zoonotic disease, allergies, and injuries, in particular in immunodeficient or immunocompromised subjects such as pediatric oncological patients or heart transplanted subjects [2,47,100]. Thus, the choice of an appropriate animal and activity program is fundamental [47,101].

As regards the fear of disease transmission, veterinarians should be more actively involved. The presence of veterinarians in the hospitals is mainly requested by the animal owner, according to hospital policies. Acquiring the veterinarian, a pivotal role in a team including hospital operators, could improve the focal point on the wellbeing of animals, mitigating the possible spread of disease and enhancing the benefit of the AAIs [102,103]. In addition, veterinarians participating in AAI sessions could also facilitate relations with hospital administration and staff and the AAI operators in the development of procedures finalized to reduce the risk and improve benefits to humans and safeguard the welfare of the “animal therapists”. Thus, screening and vaccinations should be discussed, for the zoonosis associated with any animal species involved in the AAI with veterinarians. Moreover, veterinarians must be strictly involved with the staff performing clinical protocols for AAI in both oncologic and care settings according to a One Health approach [2]. This vision is in line with the International Association of Human-Animal Interaction Organizations (IAHAIO) AAI guidelines and constitutes the maximum expression of circular health according to the One Health principle [28,104,105,106,107]. On this basis, we could realize that AAIs represent a wide range of relational feedbacks characterized first by bodily attitudes and subsequently by sensory–motor–emotional patterns between the two species. Although different animal species are involved in AAIs [25,108,109], dogs are preferred because they seem to facilitate a greater mutual relationship and deeper therapeutic work [109]. Moreover, for its ethological features such as relationship, communication and interaction, and similarities with the children, the dog learns through games [110]. In fact, dogs are considered the most well-established animals involved in AAIs directly in hospitals, daycare centers, or homes for the elderly offering many advantages, particularly for those clients affected by mental disorders or forced to stay in those facilities for a long time [111,112,113]. Involving dogs in AAIs has several benefits that are greater [114,115,116] than the risks. Some find potential stress or welfare risks in dogs occur that might limit or even avoid the involvement of dogs in the therapies. Generally, small dogs are certainly the most well-established AAI. Thus, a careful evaluation of the dogs employed of different sizes and breeds in an AAI program would be necessary [100].

The purpose of the different types of AAIs is the planning of games that the handler, with a practiced doctor or psychologist, conducts on the basis of the animal species involved, and its individual characteristics.

A limit of the present review deals with the few data describing zoonosis, habits, advantages, and regulations managing the employment of the numerous nonofficial or nonconventional animal species in the AAIs in different countries. Thus, an official list of the specialized centers and recognized structures, professionals involved, AAIs projects, and identification number of the AAI animals involved, in each country, would improve the quality level of AAI programs. Such lists should be managed by the regional/national authorities, in order to guarantee the qualifications of the professionals. This program just started in Italy [30]. Thus, national institutions, by determining boundaries and providing indications for the correct implementation of AAIs, would assure protection for both the humans and animals involved enhancing the safety and effectiveness of the interventions, and their overall quality [30].

## 7. Future Directions

The synergy among veterinarians, public health professionals, and epidemiologists, has a key role in preventing zoonotic disease transmission to safeguard the health of humans, animals, and the environment, in accordance with a One Health vision. AAIs, and more specifically the AAT, and the AAA in the health sector, constitute a tangible representation of the One Health perspective and therefore, necessitate a multidisciplinary/intersectoral approach between the different health professional figures who, each according to their own skills, work in a specialized team for the prevention and control of zoonoses, the health and welfare of people, the animals involved, and the environment.

Furthermore, we would build an action plan among different international experts in the field of AAI in order to formulate standardized hygiene–health–behavioral procedures, with the ultimate aim of establishing univocal health and behavioral certifications for all animals performing AAT and AAA in the healthcare field.

## 8. Conclusions

In conclusion, since zoonoses are considered public health problems, the use of a multidisciplinary and integrated One Health approach is necessary for their resolution. This strategy provides for a series of synergies aimed at surveillance, monitoring, and prevention, as well as the improvement of training/information/dissemination. As described in the literature, AATs are finalized for the treatment of various types of pathologies and with a state of immunocompromise or immunosuppression. Veterinarians’ involvement in these programs is crucial because they serve the health and welfare of animals participating in these programs and as experts in preventing zoonotic disease transmission. On this basis, we might hypothesize that a correct procedural protocol including a preventive, integrated, and multidisciplinary approach, will represent the future development of AAIs. Furthermore, a benefit/challenge analysis carried out in this review allows us to hypothesize that, by applying preventive measures, AAIs could be considered as cotherapy for specific pathologies and safe under the hygienic-sanitary point of view for the patients treated, for the operators, and for the animals involved.

## Figures and Tables

**Table 1 animals-13-01592-t001:** Search string used in PubMed, Web of Science, SCOPUS, and Google Scholar.

(1) “animal-assisted therapy” OR “animal assisted interventions” AND “zoonosis”
(2)“animal-assisted interventions” OR “animal-assisted therapy” AND “dog” AND “zoonosis”
(3) “animal-assisted interventions” OR “animal assisted therapy” AND cat AND “zoonosis”
(4) “animal-assisted interventions” OR “animal-assisted therapy” AND “rabbit” AND “zoonosis”
(5) “animal-assisted interventions” OR “animal-assisted therapy” AND “equine” AND “zoonosis”

## Data Availability

Not applicable.

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
