# Peer review of "Human-Animal Interaction in Animal-Assisted Interventions (AAI)s: Zoonosis Risks, Benefits, and Future Directions—A One Health Approach"

_animals, 2023, doi:10.3390/ani13101592_

Round 1
Reviewer 1 Report
I found this to be an interesting manuscript, and I think it does have considerable potential to make a significant contribution to the literature. However, I also think there are some important issues with the manuscript, in its current form, which limits the contribution it makes. Most notably, the lack of methodological detail. Specific comments below:
Lines 98-101 outline the ‘aims’ of the review but are quite vague. I’d advise the exact research questions to be written out at the end of the introduction.
The authors state on line 102 that three electronic databases have been consulted, but only list PubMed and Web of Science. What was the third database?
The authors indicate they used 5 search strings (line 104-line 105), please could the authors indicate how these were developed? I assume via searching databases with subject headings/free-text words that relate to your research topic.
There is very little detail about the design of the review – did the authors do a narrative, scoping, or systematic review? Please clarify. Ideally, the authors require a methods section with much more detail (especially if this was a systematic search), including the data analysis plan. Please do provide a much more detailed account of the analysis plan.
Results:
I’d expect to see a brief summary section at the beginning of the results to summarise how many papers were included in the review and a brief overview of their characteristics (e.g., study design, intervention type).
Lines 153 – 156: The authors state ‘another problem that can be found in AAIs is the patient or family’s fear of dogs’. There is a lot of existing research that indicates fear of dogs/allergies is in fact an exclusion criterion, so participants would not be coerced into receiving an intervention that would actually have a negative impact on them. I think it’s important to include this counter point.
The authors also state that ‘generally, small dogs are preferred’ with one citation. Preference of dog sizes and breeds vary significantly across individuals, and AAIs can involve a huge variation dependent on individual preferences and the overarching aims of the intervention. I agree with lines 154-156 when they highlight the need for careful evaluation of dogs of different sizes and breeds, though.
Line 182: a full stop after allergy that shouldn’t be there?
I wonder why the authors have grouped cats and rabbits together when there is a lot of information available for cats, and cat-assisted interventions (while lesser used than dog-assisted interventions) are still popular in their own right. I think it would be worth separating cat/rabbits and providing an introductory statement to rabbit-assisted interventions clarifying that less studies were available to extract data from.
Section 4: equine-assisted interventions: The description of EAIs is valuable, but would be better placed in the introduction, as it does not fit under the heading ‘potential zoonoses associated with equine assisted interventions’, nor do the authors provide descriptions of the other AAIs in the sections above.
Lines 247-250: Please provide a reference for these sentences in relation to donkey’s behaviour.
Line 270: I would advise to use the term ‘animals’ rather than ‘pets’ (also line 290). While the animal may be the handler’s pet, they are not the participants pet. I’d also suggest to change the word ‘using’ to ‘involving’, as the authors insinuate the animal is being ‘used’ in the intervention as opposed to being an active and important element in the intervention delivery.
Line 304: ‘Dogs are preferred’ is a very over generalised statement based on one reference – they are certainly the most well-established AAI, but not necessarily preferred.
Author Response
Dear Editor,
we appreciated very much comments and suggestions from you and the reviewers (#1, #2 and 3#) in respect to our Review [Animals] Manuscript ID: animals-2339225. As kindly suggested, we have provided to revise the overlap of our manuscript (part marked 1, 2 in the iThenticate report) highlighted in red within the main body of the text.
In this regard, Reviewer’s comments have allowed us to significantly improve our manuscript.
Please find below a point-by-point reply to the Reviewers’ comments (reported in bold).
As requested, we used for the revision the file downloaded where the major revisions are highlighted in red within the main body of the text.
We hope that the current revision could be suitable for the publication in Animals.
Regards,
Giovanna Liguori, Anna Costagliola, Renato Lombardi, Orlando Paciello and Antonio Giordano.
Author's Notes to Reviewer 1
We wish to thank you for the time spent on our manuscript (Paper Animals -2339225) and for the opportunity to submit a revised version of the manuscript modified in response to your comments.
My co-authors and I modified several sentences in the manuscript accordingly. Below you will find a point-by-point answer to all remarks and changes we made in the manuscript. The changes in the sentences of the manuscript were made by ticking the original text downloaded and by adding the new text. Every change was highlighted in red.
In Bold are reported the Reviewers comments.
Comments of the Reviewer #1:
Comments and Suggestions for Authors
I found this to be an interesting manuscript, and I think it does have considerable potential to make a significant contribution to the literature. However, I also think there are some important issues with the manuscript, in its current form, which limits the contribution it makes. Most notably, the lack of methodological detail. Specific comments below:
lines 98-101 outline the ‘aims’ of the review but are quite vague. I’d advise the exact research questions to be written out at the end of the introduction:
answer: We thank the reviewer for his/her/punctual observation and we have provided to modified as follows: “The present narrative review focuses on the most common zoonotic agents that can be transmitted from animals to human and the ways of preventing them, to be applied both to the animals and to the patient; in choosing the right animal species according to the single patient needs and linked that might occur during AAIs accordingly to the Italian National Guidelines for AAIs 2015 (Italian National Guidelines for Animal Assisted Interventions (AAI). 2015). For this purpose, these guidelines include: a) specific AAI training for each figure involved in AAI (veterinarian, animal handler, etc.); b) the establishment of a regional register of traders and facilities; c) health, welfare and behavioral criteria for the animals involved; d) the scientific evaluation of the results of projects performed”.
The authors state on line 102 that three electronic databases have been consulted, but only list PubMed and Web of Science. What was the third database?
answer: We appreciate the suggestion of the reviewer and we have added the other electronic databases after the revision procedure Scopus and Google Scholar.
The authors indicate they used 5 search strings (line 104-line 105), please could the authors indicate how these were developed? I assume via searching databases with subject headings/free-text words that relate to your research topic.
answer: As rightly indicated by the reviewer, we have provided to modify the period as follows: “using five strings that represent subject heading and text words related to the our topic”.
There is very little detail about the design of the review – did the authors do a narrative, scoping, or systematic review? Please clarify. Ideally, the authors require a methods section with much more detail (especially if this was a systematic search), including the data analysis plan. Please do provide a much more detailed account of the analysis plan.
answer: As kindly suggested, we have clarified in the Introduction section that our manuscript is a narrative review.
Results:
I’d expect to see a brief summary section at the beginning of the results to summarise how many papers were included in the review and a brief overview of their characteristics (e.g., study design, intervention type)
answer: As suggested by the reviewer that we thank for his/her punctual attention, we have clarified in the Introduction section that our manuscript is a narrative review.
Lines 153 – 156: The authors state ‘another problem that can be found in AAIs is the patient or family’s fear of dogs’. There is a lot of existing research that indicates fear of dogs/allergies is in fact an exclusion criterion, so participants would not be coerced into receiving an intervention that would actually have a negative impact on them. I think it’s important to include this counter point.
answer: As suggested by the reviewer, we have provided to add the following sentences in the text and the relative references:“Another problem that can be found in the AAIs is the patient or family’s fear of dogs: one of exclusion criterion of an animal involved in AAI, in such circumstances. In fact, the exclusion criteria can include: - fear/phobia of animals, cultural attitudes (Chur-Hansen et al, 2014); -unsafe animal behavior (Friesen, 2010); - injuries, such as a fall or bites and scratches, can happen if handled inappropriately or an inappropriate animal for the therapy environment is chosen (Brodie et al., 2002; Bert et al., 2016); -allergic reactions such as pet dander (Schmidt et al., 2022); -a well as concerns regarding hygiene/sanitization (Schmidt et al, 2022). Most protocols suggest that dogs undergo washing and grooming by 24 h prior to contact with patients (Lefebvre et al, 2008b). Nevertheless, washing a dog a couple of times during a week can reduce Can f1 from dog hair and coat lowering the risk of allergic reaction (Bert et al., 2016). Allergies can cause skin irritations, rhinoconjunctivitis, and bronchial asthma, representing a precise contraindication for (in)direct contact with dogs (Bert et al., 2016). Up to now there is no evidence of the presence of hypoallergenic dog breve (Schmidt et al., 2022). Then, under a welfare point of view, the allergological risks of implementing dogs in a hospital or outpatient setting should not be underestimated (Schmidt et al, 2022) and that these animals should not interact with patients suffering of allergic asthma. In addition, emergency medication should be available to a trained person on site, to mitigate risk, in the event of an allergic response (Toshihiro et a.l, 2005; Schmidt et al., 2022)”.
References added:
- Bert F, Gualano MR, Camussi E, Pieve G, Voglino G, Siliquini R. Animal assisted intervention: A systematic review of benefits and risks. Eur J Integr Med. 2016 Oct;8(5):695-706.
- Brodie SJ, Biley FC, Shewring M. An exploration of the potential risks associated with using pet therapy in healthcare settings. J Clin Nurs. 2002 Jul;11(4):444-56.
- Chur-Hansen A, McArthur M, Winefield H, Hanieh E, Hazel S. Animal assisted interventions in children’s hospitals: a critical review of the literature. Anthrozoös. (2014) 27:5–18.
- Friesen L. Exploring animal-assisted programs with children in school and therapeutic contexts. Early Child Educ J. (2010) 37:261–7.
- Lefebvre SL, Golab GC, Christensen E, Castrodale L, Aureden K, Bialachowski A, et al. Guidelines for animal-assisted interventions in health care facilities. Am J Infect Control. (2008b).
- Schmidt V, Mokrá M, Demolli P, Brüggen MC, Möhrenschlager M. Allergologic pitfalls in animal-assisted interventions. Allergo J Int. (2022) 31:1–3.
- Toshihiro S,Matsui T, Suzuki K, Chida K. Effect of pet removal on pet allergic asthma. Chest. (2005) 127:1565–71.
The authors also state that ‘generally, small dogs are preferred’ with one citation. Preference of dog sizes and breeds vary significantly across individuals, and AAIs can involve a huge variation dependent on individual preferences and the overarching aims of the intervention. I agree with lines 154-156 when they highlight the need for careful evaluation of dogs of different sizes and breeds, though.
answer: As indicated by the reviewer, we have provided to elucidate the sentence in the Discussion section as follows:”In fact, dogs are considered the most well-established animal involved in AAIs directly in hospitals, daycare centers, or homes for the elderly offering many advantages particularly for those clients affected by mental disorders or forced to stay in those facilities for long time (Dimitrijevi´c, 2009; Hediger et al., 2019). Involving dogs in the AAIs has several benefits that are greater (Meers et al, 2022; Machová et al, 2019a,b) than the risks. Some find potential stress or welfare risks in dogs occur that might limit or even avoid the involvement of dogs in the therapies. Generally, small dogs are certainly the most well-established AAI. Thus, a careful evaluation of the dogs employed of different sizes and breeds in an AAI program, would be necessary”.
References added:
- Dimitrijevi´c, I. Animal-Assisted Therapy—A New Trend In The Treatment of Children and Adults. Psychiatr. Danub. 2009, 21, 236–241.
-
- Hediger, K.; Thommen, S.; Wagner, C.; Gaab, J.; Hund-Georgiadis, M. Effects of animal-assisted therapy on social behaviour in patients with acquired brain injury: A randomised controlled trial. Sci. Rep. 2019, 9, 5831.
-
- Machová K, Procházková R, Eretová P, Svobodová I, Kotík I. Effect of Animal-Assisted Therapy on Patients in the Department of Long-Term Care: A Pilot Study. Int J Environ Res Public Health. 2019a Apr 16;16(8):1362.
-
- Machová K, Součková M, Procházková R, Vaníčková Z, Mezian K. Canine-Assisted Therapy Improves Well-Being in Nurses. Int J Environ Res Public Health. 2019b Sep 30;16(19):3670.
-
- Meers LL, Contalbrigo L, Samuels WE, Duarte-Gan C, Berckmans D, Laufer SJ, Stevens VA, Walsh EA, Normando S. Canine-Assisted Interventions and the Relevance of Welfare Assessments for Human Health, and Transmission of Zoonosis: A Literature Review. Front Vet Sci. 2022 Jun 17;9:899889. doi: 10.3389/fvets.2022.899889.
line 182: a full stop after allergy that shouldn’t be there?
answer: We agree perfectly with the reviewer and regret for this inaccuracy. The full stop has been removed
I wonder why the authors have grouped cats and rabbits together when there is a lot of information available for cats, and cat-assisted interventions (while lesser used than dog-assisted interventions) are still popular in their own right. I think it would be worth separating cat/rabbits and providing an introductory statement to rabbit-assisted interventions clarifying that less studies were available to extract data from.
answer: Accordingly, to the suggestion of the reviewer that we thank for helping us to improve the quality of the manuscript, the section cat/rabbit has been separated in two different cat and rabbit. A short introduction to rabbit paragraph follows has been introduced, as follows:“The literature referring to this species is almost scarce and only few data were possible to be extracted (see Table 2).”.
Section 4: equine-assisted interventions: The description of EAIs is valuable, but would be better placed in the introduction, as it does not fit under the heading ‘potential zoonoses associated with equine assisted interventions’, nor do the authors provide descriptions of the other AAIs in the sections above..
answer: We perfectly agree with the reviewer regret for our inattention. We have provided to place the description of EAIs in the Introduction section.
Lines 247-250: Please provide a reference for these sentences in relation to donkey’s behaviour.
answer: We perfectly agree with the reviewer and we have provided to insert the following references for the sentences in relation to donkey’s behavior:
- McLean AK, Navas González FJ, Canisso IF. Donkey and Mule Behavior. Vet Clin North Am Equine Pract 2019, 35:575-588.
- Gonzalez-De Cara, C.A.; Perez-Ecijaa, A.; Aguilera-Aguilera, R.; Rodero-Serrano, E.; Mendoza, F.J. Temperament test for donkeys to be used in assisted therapy. Appl Anim Behav Sci 2017, 186, 64-71.
Line 270: I would advise to use the term ‘animals’ rather than ‘pets’ (also line 290). While the animal may be the handler’s pet, they are not the participants pet. I’d also suggest to change the word ‘using’ to ‘involving’, as the authors insinuate the animal is being ‘used’ in the intervention as opposed to being an active and important element in the intervention delivery.
answer: As suggested the word “pets” has been replaced with “animals”. The terms using/use were substitute with involving/involvement, respectively.
Line 304: ‘Dogs are preferred’ is a very over generalised statement based on one reference – they are certainly the most well-established AAI, but not necessarily preferred.
answer: As suggested above we have provided to vary the sentences as follows:”In fact, dogs are considered the most well-established animal involved in AAIs directly in hospitals, daycare centers, or homes for the elderly offering many advantages particularly for those clients affected by mental disorders or forced to stay in those facilities for long time (Dimitrijevi´c, 2009; Hediger et al., 2019). Involving dogs in the AAIs has several benefits that are greater (Meers et al, 2022; Machová et al, 2019a,b) than the risks. Some find potential stress or welfare risks in dogs occur that might limit or even avoid the involvement of dogs in the therapies. Generally, small dogs are certainly the most well-established AAI. Thus, a careful evaluation of the dogs employed of different sizes and breeds in an AAI program, would be necessary”.
References added:
-
- Dimitrijevi´c, I. Animal-Assisted Therapy—A New Trend In The Treatment of Children and Adults. Psychiatr. Danub. 2009, 21, 236–241.
-
- Hediger, K.; Thommen, S.; Wagner, C.; Gaab, J.; Hund-Georgiadis, M. Effects of animal-assisted therapy on social behaviour in patients with acquired brain injury: A randomised controlled trial. Sci. Rep. 2019, 9, 5831.
-
- Machová K, Procházková R, Eretová P, Svobodová I, Kotík I. Effect of Animal-Assisted Therapy on Patients in the Department of Long-Term Care: A Pilot Study. Int J Environ Res Public Health. 2019a Apr 16;16(8):1362.
-
- Machová K, Součková M, Procházková R, Vaníčková Z, Mezian K. Canine-Assisted Therapy Improves Well-Being in Nurses. Int J Environ Res Public Health. 2019b Sep 30;16(19):3670.
- Meers LL, Contalbrigo L, Samuels WE, Duarte-Gan C, Berckmans D, Laufer SJ, Stevens VA, Walsh EA, Normando S. Canine-Assisted Interventions and the Relevance of Welfare Assessments for Human Health, and Transmission of Zoonosis: A Literature Review. Front Vet Sci. 2022 Jun 17;9:899889.

Reviewer 2 Report
This article seems to be a narrative review of zoonotic agents and practice in AAIs. In the title, the authors referred to a "benefit/challenges analysis" that is not clearly described in the paper therefore I suggest the authors change the title or develop this analysis using scientific methods.
Introduction:
line 75-80 I suggest this paper which summarizes the contents of Italian National Guidelines: Simontato et al., 2018 "The Italian Agreement between the Government and the Regional Authorities: National Guidelines for AAI and Institutional Context" People& Animals, 1(1).
Line 85-88: this is not properly reported, dogs, cats, rabbits, horses and donkeys can be involved in AAT and AAE, whereas all domestic species can be involved in AAA. Moreover, providers can involve domestic species other than those allowed for AAT and AAE in these two types of interventions obtaining a positive assessment of their project by the NRC for AAI and the Italian Ministry of Health.
Line 95: change AAT and AAA with AAI
Line 97: please introduce a reference to support your sentence
Line 99-104: define better the aim of your review, I suppose this is a narrative review, anyway methods applied for search strategy and its results need to be deepened.
Line 121: Norovirus?
Line 122 and Line 124: there is a repetition
Line 130-132: add a reference
Line 170-184: you can maybe introduce a sentence about patient selection
Line 185-186: add a reference
Add a paragraph about the limitations of the study in the discussion
Author Response
Dear Editor,
we appreciated very much comments and suggestions from you and the reviewers (#1, #2 and 3#) in respect to our Review [Animals] Manuscript ID: animals-2339225. As kindly suggested, we have provided to revise the overlap of our manuscript (part marked 1, 2 in the iThenticate report) highlighted in red within the main body of the text.
In this regard, Reviewer’s comments have allowed us to significantly improve our manuscript.
Please find below a point-by-point reply to the Reviewers’ comments (reported in bold).
As requested, we used for the revision the file downloaded where the major revisions are highlighted in red within the main body of the text.
We hope that the current revision could be suitable for the publication in Animals.
Regards,
Giovanna Liguori, Anna Costagliola, Renato Lombardi, Orlando Paciello and Antonio Giordano.
Author's Notes to Reviewer 2
We wish to thank you for the time spent on our manuscript (Paper Animals -2339225) and for the opportunity to submit a revised version of the manuscript modified in response to your comments.
My co-authors and I modified several sentences in the manuscript accordingly. Below you will find a point-by-point answer to all remarks and changes we made in the manuscript. The changes in the sentences of the manuscript were made by ticking the original text downloaded and by adding the new text. Every change was highlighted in red.
In Bold are reported the Reviewers comments.
Comments of the Reviewer #2:
This article seems to be a narrative review of zoonotic agents and practice in AAIs. In the title, the authors referred to a "benefit/challenges analysis" that is not clearly described in the paper therefore I suggest the authors change the title or develop this analysis using scientific methods.
answer: We thank the reviewer for his/her/punctual observation and we have provided to modified the title as follows: “Human-animal interaction in the Animal Assisted Interventions (AAI)s: zoonosis risks, benefits and future directions. A One Health approach”.
Introduction:
line 75-80 I suggest this paper which summarizes the contents of Italian National Guidelines: Simontato et al., 2018 "The Italian Agreement between the Government and the Regional Authorities: National Guidelines for AAI and Institutional Context" People& Animals, 1(1).
answer: We appreciate the suggestion of the reviewer and we have added the following reference: Simonato, Martina; De Santis, Marta; Contalbrigo, Laura; Benedetti, Daniele; Finocchi Mahne, Elisabetta; Santucci, Vincenzo Ugo; Borrello, Silvio; and Farina, Luca (2018) "The Italian Agreement between the Government and the Regional Authorities: National Guidelines for AAI and Institutional Context," People and Animals: The International Journal of Research and Practice: Vol. 1 : Iss. 1, Article 1.
Line 85-88: this is not properly reported, dogs, cats, rabbits, horses and donkeys can be involved in AAT and AAE, whereas all domestic species can be involved in AAA. Moreover, providers can involve domestic species other than those allowed for AAT and AAE in these two types of interventions obtaining a positive assessment of their project by the NRC for AAI and the Italian Ministry of Health.
answer: As rightly indicated by the reviewer, we have provided to modify the period as follows: “In addition to dogs, cats, rabbits, horses and donkeys which can be involved in AAT and AAE, there are other domestic animal species that are involved in AAA as well (guinea pigs, chicken, goats, ferrets etc), although they still need to be evaluated by national authorities as regards their safety and welfare before they can become suitable officially for AAT and AAE [21].”
Line 95: change AAT and AAA with AAI
answer: As rightly indicated by the reviewer, AAT and AAA have been replaced by AAI.
Line 97: please introduce a reference to support your sentence
answer: The requested reference has been added: Zoonoses - World Health Organization (WHO). 29 July 2020 https://www.who.int/news-room/fact-sheets/detail/zoonoses.
Line 99-104: define better the aim of your review, I suppose this is a narrative review, anyway methods applied for search strategy and its results need to be deepened.
answer: It is a narrative review and has already been introduced, as suggested.
Line 121: Norovirus?
answer: We agree perfectly with the reviewer and regret for this inaccuracy. It is Norovirus and the following reference has been added both in the text and table: Charoenkul K, Nasamran C, Janetanakit T, Tangwangvivat R, Bunpapong N, Boonyapisitsopa S, et al. Human Norovirus Infection in Dogs, Thailand. Emerg Infect Dis. 2020;26(2):350-353.
Line 122 and Line 124: there is a repetition
answer: As suggested, the repetition has been removed.
Line 130-132: add a reference
answer: As kindly suggested, the reference has been introduced before the full stop.
Line 170-184: you can maybe introduce a sentence about patient selection
answer: As kindly suggested, the following sentence has been added: “…or in extreme cases choosing a different animal species for the therapy”.
Add a paragraph about the limitations of the study in the discussion
answer: We thank the reviewer for his/her/punctual observation and we have provided to The following sentences about the limitations of the study in the Discussion section:” A limit of the present review deals with the few data describing zoonosis, habits, advantages and regulation managing the employment of the numerous non official or non-conventional animal species in the AAIs in different countries. Thus, an official list of the specialized centers and recognized structures, professionals involved, AAIs projects and identification number of the AAI animals involved, in each country, would improve the quality level of AAI programs. Such lists should be managed by the regional/national Authorities, in order to guarantee on the qualifications of the professionals. This program is just started in Italy (Simonato et al, 2018). Thus, national institutions by determining boundaries and providing indications for the correct implementation of AAIs, would assure protection for both humans and animals involved enhancing the safety and effectiveness of the interventions, and their overall quality (Simonato et al, 2018)”.

Reviewer 3 Report
Good manuscript. Very much relate to zoonoses in HAI is feeding raw diets to assistance animals. Most providers require that raw food not be use for at least 30 to 90 days before service. This should be addressed.
In addition, the paper could have addressed animal bites, slips, and falls. While not zoonoses, are a very real problem so should be acknowledged.
Author Response
Dear Editor,
we appreciated the evaluation of the reviewers #1, #2 and 3# in respect to our Review [Animals] Manuscript ID: animals-2339225. As kindly suggested, we have provided to revise the overlap of our manuscript (part marked 1, 2 in the iThenticate report) highlighted in red within the main body of the text.
In this regard, Reviewer’s comments have allowed us to significantly improve our manuscript.
Please find below a point-by-point reply to the Reviewers’ comments (reported in bold).
As requested, we used for the revision the file downloaded where the major revisions are highlighted in red within the main body of the text.
We hope that the current revision could be suitable for the publication in Animals.
Regards,
Giovanna Liguori, Anna Costagliola, Renato Lombardi, Orlando Paciello and Antonio Giordano.
Author's Notes to Reviewer 3
We wish to thank you for the time spent on our manuscript (Paper Animals -2339225) and for the opportunity to submit a revised version of the manuscript modified in response to your comments.
My co-authors and I modified several sentences in the manuscript accordingly. Below you will find a point-by-point answer to all remarks and changes we made in the manuscript. The changes in the sentences of the manuscript were made by ticking the original text downloaded and by adding the new text. Every change was highlighted in red.
In Bold are reported the Reviewers comments.
Comments of the Reviewer #3:
Good manuscript. Very much relate to zoonoses in HAI is feeding raw diets to assistance animals. Most providers require that raw food not be use for at least 30 to 90 days before service. This should be addressed.
answer: We thank the Reviewer for his/her very careful reading of our manuscript and for his/her important comments. As kindly suggested, the following sentence and the relative references were added:” Dogs involving in AAIs should not be fed raw animal-origin foods within 90 days before service (Lefebvre et al. 2008; Murthy et al., 2015)
References added:
-
Murthy R, Bearman G, Brown S, Bryant K, Chinn R, Hewlett A et al (2015) Animals in healthcare facilities: recommendations to minimize potential risks. Infect Control Hosp Epidemiol 36 (5):495–516.
-
Lefebvre, S.L.; Peregrine, A.S.; Golab, G.C.; Gumley, N.R.; Waltner-Toews, D.; Weese, J.S. A veterinary perspective on the recently published guidelines for animal-assisted interventions in health-care facilities. J Am Vet Med Assoc 2008, 233, 394-402.
In addition, the paper could have addressed animal bites, slips, and falls. While not zoonoses, are a very real problem so should be acknowledged.
answer: We appreciate the suggestion of the reviewer and we have provided to add the following sentences to the Discussion section and the relative references below: “Bites, slisps, and falls from animals, dogs and, to a lesser extent, cats might be the most troublesome animal-associated health hazard (Vines, 1993) in terms of seriousness, frequency and cost. Moreover, the most troublesome breeds, the importance of a good temperament and the need for schooling could be incorporated into guidelines to reduce the risk of injuries in a AAI program. It is reasonable to suggest that, in a well-supervised environment such as a hospital ward, after careful selection of the AAI animal involved, the risks of animal bites are minimal and should not prevent the implementation of such therapy. In fact, animal related accidents have to be considered very rare although they can happen. For that appropriate reviews and guidelines implement all security precautions effectively minimizing risks (Brodie et al., 2002; Bert et al., 2016)”.
References added:
-
Bert F, Gualano MR, Camussi E, Pieve G, Voglino G, Siliquini R. Animal assisted intervention: A systematic review of benefits and risks. Eur J Integr Med. 2016 Oct;8(5):695-706.
-
Brodie SJ, Biley FC, Shewring M. An exploration of the potential risks associated with using pet therapy in healthcare settings. J Clin Nurs. 2002 Jul;11(4):444-56.
-
Vines G. (1993) The secret power of pets.Nursing 6, 30–34.
